# Will Quantitative Proteomics Redefine Some of the Key Concepts in Skeletal Muscle Physiology?

**DOI:** 10.3390/proteomes4010002

**Published:** 2016-01-11

**Authors:** Agnieszka Gizak, Dariusz Rakus

**Affiliations:** Department of Animal Molecular Physiology, Wroclaw University, Cybulskiego 30, 50-205 Wroclaw, Poland; dariusz.rakus@uni.wroc.pl

**Keywords:** C2C12 cells, kcat, glyconeogenesis, glycolysis regulation, lysine acetylation, skeletal muscle

## Abstract

Molecular and cellular biology methodology is traditionally based on the reasoning called “the mechanistic explanation”. In practice, this means identifying and selecting correlations between biological processes which result from our manipulation of a biological system. In theory, a successful application of this approach requires precise knowledge about all parameters of a studied system. However, in practice, due to the systems’ complexity, this requirement is rarely, if ever, accomplished. Typically, it is limited to a quantitative or semi-quantitative measurements of selected parameters (e.g., concentrations of some metabolites), and a qualitative or semi-quantitative description of expression/post-translational modifications changes within selected proteins. A quantitative proteomics approach gives a possibility of quantitative characterization of the entire proteome of a biological system, in the context of the titer of proteins as well as their post-translational modifications. This enables not only more accurate testing of novel hypotheses but also provides tools that can be used to verify some of the most fundamental dogmas of modern biology. In this short review, we discuss some of the consequences of using quantitative proteomics to verify several key concepts in skeletal muscle physiology.

## 1. Introduction

Traditional approaches to the identification of the role of signaling pathways in regulating cellular behavior most often focus on single components of a studied pathway, without taking into account the full complexity of the pathway (or its relationship with other pathways), let alone the complexity of a whole cell.

Often, within this pathway, the quantity (or activity) of a studied protein is not referred to quantities (or activities) of other proteins in the same pathway*,* or it is just compared under different treatments, or even between different types of cells or tissues. Such investigations tend to overstate the physiological significance of the observed changes and the protein’s regulatory role, or just provide protein ratios without interpretation related to physiological processes in a cell or tissue. While it is important to understand the physiological relevance of the changes in protein levels from one condition to another and from one cell type to another, this simply cannot be achieved if the metabolic context of the cell is overlooked.

Focusing on a small number of proteins, usually within one or two pathways, and neglecting other possible interaction partners is not likely to aid the discovery of effective treatments of diseases. To achieve this goal, we have to get the all-encompassing view of a cell’s metabolism, and for this, we need to know the full proteome of the cell. In other words, to treat a pathology we must first know what a “norm” is.

Mass spectrometry (MS)-based proteomics is currently one of the most powerful tools available to help us achieve this goal.

High-resolution proteomics combined with computational analysis can give us an in-depth view into cells’ and organelles’ proteomes, and a system-wide insight into the organization and relationships between biological processes in both health and disease. The recently developed total protein approach (TPA) provides an easy way to study proteomes quantitatively, allowing for the calculation of concentrations of proteins in any protein mixture. This approach requires a minimal amount of samples and only little pre-fractionation, making analysis more time- and resource-efficient [1,2]. Recent evidence also suggests that there is a correlation between enzyme titer and its maximal activity, indicating that enzyme concentration is a good proxy for its catalytic potential *in vivo* [3].

In this short review, we discuss the consequences of using a label- and standard-free quantitative proteomics approach to verify several key concepts in skeletal muscle physiology. Quantitative proteomics data obtained with this method correlates extremely well with our existing knowledge of the amounts of contractile proteins in striated muscle and the muscle capacity to metabolize glucose in glycolysis and produce energy in the mitochondria [4,5,6,7,8]. However, the new data also casts doubt on the correctness of some of the canonical regulatory mechanisms in striated muscle. We conclude that a thorough re-examination of these mechanisms may be necessary before we can profoundly understand muscle physiology and discover new targets for the treatment of muscle disease.

## 2. Classical Studies of Skeletal Muscle Metabolism

Beside liver, skeletal muscles belong to the most intensely studied tissues in the context of metabolism and protein composition. A framework of muscle metabolism has been established in the late 1970s of the last century on the basis of physiological, histochemical and enzymological studies (e.g., [9,10,11,12]).

The common effort of several research groups has allowed for the identification of heterogeneous fiber composition of skeletal muscles and the description of their basic biochemical and physiological properties. Based on these analyses, muscle fibers have been broadly classified into slow (type I) and fast (type II) fibers. Whereas in muscles composed mainly of slow fibers oxidative phosphorylation is the major source of ATP, in muscles built predominantly of fast fibers ATP stems mainly from glycolysis [12]. Traditional biochemical and physiological techniques have delivered several essential findings on muscle biology, such as the discovery of glycolysis regulation by fructose-2,6-bisphosphate [13,14], the description of mechanisms of glucose transport into myofibers [15], and many others. However, up to this century, all conclusions on muscle biology were based on simultaneous analysis of the activity, cellular localization and/or biological functions of only a limited number of enzymes or proteins.

The development of mass spectrometry techniques in the late 1990s allowed for the application of MS-based methods for identification and relative quantitation of thousands of proteins from complex biological samples. This unique value of MS-based proteomics has been applied for protein profiling of muscles from various animal species, bringing information both on the composition and the relative abundance of hundreds [16,17] and later thousands [6,8,18] of proteins comprising striated muscle proteomes. Simultaneously, development of different labeling technologies enabled the monitoring of alterations in muscle proteome during aging [19,20] and the progression of muscle diseases [21,22]. Combining the MS-based proteomics with cell fractionation methods allowed for the characterization of proteomes of muscle organelles [7,23]. Recent advances in label-free techniques and applications of the Total Protein Approach [24] enabled the quantitative description of the striated muscle proteome in biochemically interpreted units, such as mol per mg of total protein, which is a prerequisite for physiological/metabolic analysis of any process in cells [6].

## 3. Phosphofructokinase and Pyruvate Kinase Are Not Universal Rate-Limiting Enzymes of Glycolysis

There is a long-standing view that phosphofructokinase (PFK) and pyruvate kinase (PK) play a regulatory role in glycolysis and can limit the maximal flux through the pathway. However, to be considered as rate-limiting, the concentration of an enzyme (or, more precisely, its maximal activity) must be significantly lower than the concentrations of other enzymes in the pathway. Surprisingly, a recent comprehensive proteomic analysis of all murine glycolytic enzymes demonstrated that low levels of PK in particular, but also PFK, were found only in the liver [3], which debunks the claim that PK and PFK are universal regulators. This is not to say that they do not play a role in the regulation of glycolysis in other tissues. PFK concentration is quite low in striated muscles, second only to hexokinase [3]. However, PFK has a very high kcat (*i.e.*, value that describes how many substrate molecules can be converted into product per unit time), which means that the total maximal activity of PFK (a product of protein titer and kcat) is much higher than the activity of aldolase, phosphoglycerate kinase, or enolase (Figure 1). The titer and kcat of PKM (the muscle isoform of PK) are some of the highest among skeletal muscle glycolytic enzymes. On balance, it is not likely that PFK and PK are responsible for regulating the glycolytic flux in skeletal muscle.

**Figure 1 proteomes-04-00002-f001:**
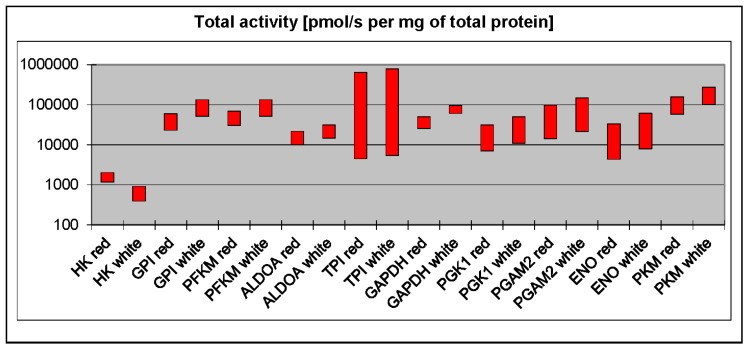
Total activities of glycolytic enzymes in red and white skeletal muscles calculated on the basis of kcat (taken from the BRENDA database) and proteomic data on the concentrations of glycolytic enzymes in skeletal muscles [6]. HK—Hexokinase; GPI—phosphoglucoisomerase; PFKM—muscle phosphofructokinase; ALDOA—aldolase A; TPI—triosephosphate isomerase; GAPDH—glyceraldehyde-3-phosphate dehydrogenase; PGK1—phosphoglycerate kinase 1; PGAM2—muscle phosphoglycerate mutase; ENO—enolase; PKM—muscle pyruvate kinase.

Another common view is that a high glycolytic capacity correlates with the ability of a tissue to synthesize a major allosteric activator of PFK—fructose-2,6-bisphosphate (F2,6P2). It is well known that fast-twitch (white) muscles have the highest capacity to degrade glucose in glycolysis and an ability to rapidly accelerate the glycolytic flux. However, proteomic studies have demonstrated that in white muscles, the titer of PFKFB (phosphofructokinase 2/fructose 2,6-bisphosphatase), the enzyme synthesizing F2,6P2, is about three times lower than in slow-twitch (red) muscles [6], which are thought to rely mostly on oxidative, non-glycolytic metabolism. Evidently, acceleration of glycolysis in white muscles does not rely on F2,6P2-related allosteric activation of PFK but, presumably, on the activity of calcium/calmodulin-dependent protein kinases, such as calcium/calmodulin-dependent kinase-2 (CAMK2).

Although calcium and CAMK2 can stimulate glycolysis by increasing the availability of glycogen-derived glucosyl units, they are not thought to directly regulate the activity of glycolytic enzymes. However, both calcium and CAMK2 have been shown to stimulate the association of glycolytic enzymes with cytoskeletal and sarcomeric proteins [25,26,27]. A number of studies indicate that such association not only alters the regulatory properties and kinetics of these enzymes [28,29], but it may also facilitate the channeling of substrates between metabolically sequential enzymes [25,29]. The hypothesis that the rate of glycolysis in striated muscles may be regulated mainly by the formation of metabolically active complexes is still not commonly accepted, but the new proteomic data showing the low probability of F2,6P-related activation of white muscle glycolysis strongly support this concept.

## 4. White Muscles Are Glyconeogenic

Although biochemical/metabolic studies have demonstrated that a significant part of glycogen in striated muscles may be derived from noncarbohydrates [30,31], muscle glyconeogenesis is thought to play a minor role in muscle energetics. However, there is recent proteomic evidence that glyconeogenesis could be a basic mechanism for replenishing glycogen stores in white muscles [6].

Mentioned above, F2,6P2 is not only an activator of glycolysis but also a potent inhibitor of fructose 1,6-bisphosphatase 2 (FBP2), a regulatory enzyme of glyconeogenesis. Proteomic studies have shown that skeletal (and, in particular, white) muscles have developed a mechanism for maintaining the F2,6P2 concentration at the lowest possible level: the concentration of TIGAR, a protein which hydrolyzes F2,6P2, is much higher than the concentration of PFKFB [6]. The low titer of PFKFB and the high titer of TIGAR in white muscles suggest that these muscles are suited to glycogen synthesis from precursors of carbohydrates, e.g., lactate. Additionally, the titer of FBP2 in white muscles is similar to that of hexokinase, the first enzyme of glycolysis and glycogen synthesis from glucose taken up from extracellular fluids. This indicates that the ability of white muscles to synthesize glycogen from lactate is not lower than from glucose.

## 5. Where Is the Lysine Acetylation?

Differences in kinetic parameters between populations of enzyme molecules may result from their post-translational modifications. Several recent studies have suggested that lysine acetylation may regulate activities of both glycolytic and gluconeogenic enzymes [32,33,34,35]. However, the experimental methods that ultimately led to this conclusion cannot estimate the ratio of modified to unmodified protein or accurately monitor the changes in abundance of acetylated peptides (e.g., Western blot; acetylated lysine-enriched proteins used as an output material for MS analyses to detect the changes) [34,35]. In a nutshell, the technique called “an enrichment” allows for partial purification of specifically modified (e.g., acetylated) proteins or peptides. Thus, as a result one should obtain only (at least in theory) modified peptides. From this, it is evident that this method is not suitable for demonstration of a proportion of modified to unmodified proteins. Analogously, the Western blot with the use of antibodies against modified proteins does not allow for the detection of total protein and, thus, does not provide any information about the modified/unmodified proteins proportion.

This is an important limitation, given that the biological significance of post-translational modifications depends not so much on their presence as on the concentration of the modified protein. None of the above-mentioned methods could provide information on the acetyl-occupancy of the analyzed sites, which means they cannot show the physiological/metabolic role of acetylation.

Our recent in-depth quantitative proteomic studies of mouse brain, liver, and skeletal muscles did not detect lysine-acetylated glycolytic enzymes, which suggests that if their acetylation is a general phenomenon, then it applies only to a very small protein fraction that escaped detection [3]. Thus, it is not likely that lysine acetylation directly affects the activity of glycolytic or gluconeogenic enzymes, contrary to what previous studies have suggested. Instead, the lysine-acetylated enzymes could modulate the rate of glycolytic/gluconeogenic flux, acting as a regulator of transcription, translation, and/or activity of other regulatory proteins.

## 6. C2C12 Cells Are a Good Model Only of Model Cells

The C2C12 cells, an immortalized murine myoblast cell line, are commonly used as a model system to study the molecular and biochemical properties of skeletal muscles [36,37]. There are, for example, more than 700 papers in the PubMed database examining the mechanisms of action of insulin in the C2C12 cells. A recent proteomic analysis has provided evidence that these cells differ significantly from skeletal muscle fibers in terms of their concentrations of metabolic enzymes and proteins involved in hormonal signal transduction [8]. To start with, in skeletal muscles, the insulin-dependent glucose transporter GLUT4 is about 30 times more abundant than GLUT1, whereas in C2C12 myotubes, GLUT1 is the predominant form. This indicates that glucose uptake by the C2C12 cells is largely independent on insulin stimulation. Conversely, all other proteins involved in insulin/IGF signaling (such as PI3K, PDK, Akt and Rab), and AMP-dependent kinases are much more abundant in the C2C12 myotubes. Based on these findings, it is evident that the C2C12 cells are not a correct model for studying the insulin-mediated glucose uptake. Actually, the C2C12 myotubes, given their approximately two-times-lower expression of glycolytic, tricarboxylic acid cycle and oxidative phosphorylation proteins than observed in skeletal muscles [8] are far from being the perfect model to study muscle metabolism at all.

## 7. Conclusions and Perspectives

Over the past decade, a great variety of proteomic approaches have been developed and they have been employed to investigate the changes in proteomes of normal muscle fibers and muscles affected by pathological conditions such as type 2 diabetes, neuromuscular disorders, age-related sarcopenia and dystrophy [21,38,39,40]. Rather than focusing on a small number of proteins, MS-based proteomics provides a comprehensive picture of the changes in total protein levels within the muscle fibers. Unfortunately, results obtained using diverse methodological approaches are often hard to compare, especially when these approaches differ significantly in the depth of proteome analysis. However, the discrepancies in numbers (and types) of detected proteins do not always arise only from methodological flaws. They might be partially explained by different motoric behavior of various mammalian species, differences in animal handling and even the use of animals from various providers.

In recent years, new MS-based techniques have enabled us to monitor the changes in protein concentrations and determine these concentrations with precision in the studied cells and tissues, without prior treatment with labeling compounds [1,2,3]. Combining these quantitative analyses with other biochemical methods has not only allowed us to reduce sample handling but also provided a new tool for verification of previously collected data. For example, a correlation between an enzyme titer and its maximal activity (Amax) has recently been described [3]. Based on this correlation a fundamental constant which describes an enzyme’s behavior, kcat, could be calculated simply by dividing Amax (measured in crude homogenate) by the enzyme concentration [3]. Kcat represents the number of reactions catalyzed by an enzyme during a time unit and is numerically equivalent to the turnover number (and expressed as (s^−1^)). Until now, the determination of kcat required time-consuming and laborious purification of enzymes prior to kinetic measurements and, as a result, kcat values have only been determined for a limited number of enzymes. Moreover, since the procedures of an enzyme isolation used across various laboratories are not standardized, they can often result in isolation of different populations of the enzyme (*i.e.*, selected by discrete ddifferences in quaternary/tertiary structure of the protein). This lack of standardization may explain why the kcat values for any enzyme determined by different laboratories vary significantly (BRENDA database, http://www.brenda-enzymes.org/). Thus, it is controversial whether a single, structurally homogeneous population of molecules is representative of the entire pool of enzyme molecules in a cell. Combining quantitative proteomics with measurements of the maximal enzymatic activity in whole tissue homogenates could provide the real average kcat value for all molecules of an enzyme in a tissue/cell. A majority of the kcat values determined using this combination of methods were found to be within the wide range determined for purified enzymes (see the BRENDA database). Hence, this simplified method might facilitate the identification of potential targets for therapy.

The data obtained using quantitative proteomics has so far confirmed some of the earlier views on muscle physiology, and it broadened our knowledge about muscles in healthy and pathological states; however, it also questioned some of the fundamental biological hypotheses about the functioning of skeletal muscles (Figure 2).

**Figure 2 proteomes-04-00002-f002:**
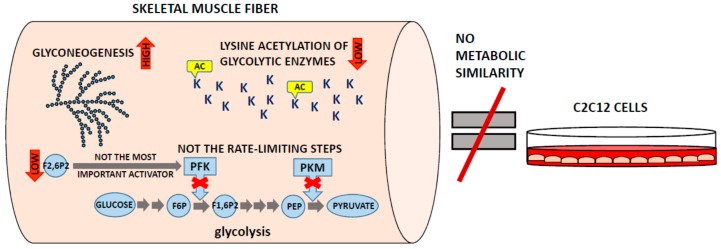
Schematic summary of the new, unexpected findings on the skeletal muscle metabolism obtained with the use of quantitative proteomics. AC—acetylation; F1,6P2—fructose-1,6-bisphosphate; F2,6P2—fructose-2,6-bisphosphate; F6P—fructose-6-phosphate; K—lysyl residues of glycolytic enzymes; PEP—phosphoenolpyruvate; PFK—phosphofructokinase; PKM—muscle isoform of pyruvate kinase.

As the next step, this emerging picture of muscle metabolism should be validated using other experimental techniques. However, should the picture prove to be accurate, proteomics promises to vastly improve our understanding of muscle physiology and aid the discovery of better targets for the treatment of muscle diseases.

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
