# Peer review of "Will Quantitative Proteomics Redefine Some of the Key Concepts in Skeletal Muscle Physiology?"

_proteomes, 2016, doi:10.3390/proteomes4010002_

Round 1

Reviewer 1 Report

The short review by Gizah et al aims to emphasize the importance of high through put proteomic approaches for advancing the field of skeletal muscle physiology.

Although the area of reseach is significant and is likely to be useful to researchers in this field; the reveiw lacks an overall flow and needs to be revised. Specific comments:

The background educate readers about the some of the pioneering work done in this field. Subsequently, the authors should focus current gaps in knowledge in the field and how proteomics would help to address these area of need. 

The authors seem to go back and forth metabolism and proteins and yet there is no section or discussion about global metabolomics approaches that could be used. The authors should also discuss what is known about resting vs exercising vs wasting muscle physiology, what are the signaling pathways including Ca2++ signaling as well as the Na and K channels that have been reported and how "omics approaches would further the field.

There should at least be some elementary discussion on different types of proteomic and metabolomic approaches that are available for use.

In the summary section, it would be good to discuss what are the drawbacks of "omics" approaches and how basic science and highthrough put approcahes complement each other. Impact of sample quality and validation of discovery findings has not been discussed.

Finally, the title should be changed since qunatitative proteomics is likely to broaden the scope and add to the knowledge rather than "Runining" the basic science contributions

Lots of typos and the general writing flow needs extensive edits

Author Response

We have carefully considered the Rewiever’s comments and we could agree that a review prepared according to the detailed plan described by the Reviewer might be of interest for a broad group of readers.

However, our short paper was not intended to be just another review about muscle physiology or available proteomic approaches and their drawbacks. Readers interested in these subjects may choose among numerous already published reviews, which are often not very different from one another. Instead, we decided to write a paper on how recent results of proteomic studies may influence some dogmas in muscle biology. Not just how proteomic (and metabolomic) approach enlarges our understanding of muscle physiology. Therefore we cannot change the paper according to the Rewiever’s suggestions.

The paper has been corrected by native English speaking scientist and we hope that it is easier to read in its present form.

Reviewer 2 Report

Major comments

Globally speaking, the topic of the review by Agnieszka Gizak and Dariusz Rakus can be considered as of quite broad interest for the scientific community. I actually believe that some of the observations raised by the authors in this manuscript really deserve consideration and diffusion. However, on the other hand, I judge the manuscript in the present form as partially confused and/or incomplete in some parts.

More specifically, I would suggest the authors to:

1) dedicate a specific paragraph to how proteomics can positively corroborate and verify data previously observed by “classical" biochemistry. In other words, the authors’ sentence (p. 2, lines 31-33) stating that “quantitative proteomics data excellently correlates with textbook knowledge on the amount of contractile proteins in striated muscles, and on capability of these muscles to metabolize glucose in glycolysis and produce energy in mitochondria” should be justified, detailed and massively expanded, as it is crucial in the “economy” and structure of this review.

2) change the title. The title is indeed not trivial in itself and somehow (and interestingly) “provoking”. However, I believe that the reader should be aware of the specific focus of the review (i.e. skeletal muscle proteomics) just by reading the title, and, at the present stage, it is definitely not the case. I would therefore suggest to add some indication about the main topic in the title.

3) move the information enclosed in the paragraph “kcat determination does not require laborious purification procedures” in a newly structured “Conclusions and perspectives” paragraph. This section contains in fact suggestions for future investigations, rather than results from previous studies (as the “core” paragraphs of a review article should do).

4) ensure that the manuscript is thoroughly reviewed by an English native speaking person. At this stage, many sentences in the manuscript are quite confused and need to be rephrased (see Minor comments as an example).

Minor comments

Page 2

Line 1: “the role” should be preferred to “a role”

Line 7: Would the authors mean something like “are prone” or “tend” when using the words “are bound”? Please clarify.

Line 8: “just to provide”

Line 20: “capable of analyzing” is the correct form

Line 29-30: “not annotated in databases” may be given into parentheses

Line 30:Would the authors mean something as “permits” or “allows” when using the term “imposes”? Please make this sentence clearer.

Line 31: “the” should be removed

Line 31-34: The exact meaning of this sentence is not totally clear to me, and should be rephrased. First, in my opinion the authors should state more clearly that (and, mostly, why) this review will deal specifically with the skeletal muscle proteome. In addition, the term “accuracy” seems not to explain the authors’ idea about the contrast between “old” biology “dogmas” and novel information achieved by proteomics; a term like “reliability” may fit better.

Line 37: I guess a word like “stating” before “that” is missing. Moreover, to the best of my knowledge, “being” should be changed into “would be” or “are”.

Line 39: “a” should be eliminated

Page 4

Line 5: I would modify the dash in “, i.e.”

Line 24: the article “a” should be added before “recent proteomic analysis”

Line 40: “PuMed” should be corrected into “PubMed” and “the” into “a”

Page 6

Line 3: I would change this sentence as follows: “It has to be remembered that, when studying post-translational modifications of a proteins, the biological”

Line 18: the abbreviation “MS” for mass spectrometry should be defined much earlier in the manuscript (e.g. page 2, line 18)

Line 22: the article “a” should be removed

Author Response

1) There are many reviews on proteomic studies corroborating "classical" knowledge of muscle physiology and describing how proteomic approach enlarges our understanding of muscle physiology published during the last several years (for example: PMID: 23777215; PMID: 20709194; PMID: 21798084; PMID: 20377394). Therefore, instead of multiplying the existent scheme of review once again, we decided to concentrate on  this part of recently obtained proteomic results which could lead to change of "classical" view of muscle physiology.

Thus, we would prefer not to extend this short manuscript to include examples of "positive verification" of data obtained by classical biochemical methods, since we feel that such data is out of scope of this manuscript, and readers may find the data in numerous already written reviews.

2) Following this suggestion we have changed the title: Will quantitative proteomics redefine some of the key concepts in skeletal muscle physiology?

We have also added indication about the main topic of the paper to the introductory part of the manuscript (in the reviewed version, all the added sentences and phrases are marked in red).

3) Following this suggestion we have included the paragraph to “Conclusions and perspectives” (text marked in red).

4) The paper has been read and corrected by native English speaking scientist and we hope that it is easier to read in its present form.

Round 2

Reviewer 2 Report

The overall level of the manuscript has been substantially improved, although a few minor typos still remain to be fixed.

Author Response

Dear Editors,

We do believe that there has been a misunderstanding, so we would like to try and clear it up. When we accepted your invitation to contribute to Invited Feature Article section of the Proteomes journal we did not realize that our article is expected to be written according to a precise vision presented to us, quite forcibly, by reviewers and editors of the journal.

In such circumstances, since we do not want to waste any more of your time, we decided to retract our submission from Proteomes.

We would like to thank you for the time dedicated to read our manuscript and we strongly believe that the scheme proposed by the reviewers will soon be developed by them into a publication which becomes a key point of reference in muscle proteomics.

Best regards,

Agnieszka Gizak